# REGULARIZED KL-DIVERGENCE FOR WELL-DEFINED FUNCTION SPACE VARIATIONAL INFERENCE IN BNNS

## ABSTRACT

Bayesian neural networks (BNN) promise to combine the predictive performance of neural networks with principled uncertainty modeling important for safety-critical applications and decision making. However, uncertainty estimates depend on the choice of prior, and finding informative priors in weight-space has proven difficult. This has motivated variational inference (VI) methods that pose priors directly on the function generated by the BNN rather than on weights. In this paper, we address a fundamental issue with such functions-space VI approaches pointed out by Burt et al. (2020), who showed that the standard objective function (ELBO) is negative infinite for many interesting priors. Our solution builds on *generalized* VI (Knoblauch et al., 2019) with the regularized KL divergence (Quang, 2019). Experiments show that our inference method accurately approximates the true Gaussian process posterior on synthetic and small real-world data sets, and provides competitive uncertainty estimates for regression and out-of-distribution detection compared to BNN baselines with both function and weight space priors.

## 1 INTRODUCTION

Neural networks have shown impressive results in many fields but fail to provide well calibrated uncertainty estimates, which are essential in applications associated with risk, such as healthcare (Kompa et al., 2021; Abdullah et al., 2022) or finance (Bew et al., 2019; Wong, 2023). Bayesian neural networks (BNNs) offer to combine the scalability and predictive performance of neural networks with principled uncertainty modeling by explicitly capturing epistemic uncertainty, a type of uncertainty that results from learning from finite data. While the choice of prior in the Bayesian framework strongly affects the uncertainty later obtained from the posterior, specifying informative priors on BNN weights has proven difficult and is hypothesized to have limited their practical applicability (Knoblauch et al., 2019; Cinquin et al., 2021; Tran et al., 2022). For instance, the default isotropic Gaussian prior, which is often chosen for tractability rather than for the beliefs it carries (Knoblauch et al., 2019), is known to have pathological behavior in some cases (Tran et al., 2022). A promising approach to solve this issue is to place priors directly on the functions generated by the BNN instead of the weights. While being technically more challenging, function-space priors allow incorporating interpretable knowledge into the inference, for instance allowing to use the extensive Gaussian Process (GP) literature to improve prior selection and design (Williams & Rasmussen, 2006).

A recent line of work has focused on using function-space priors in BNNs with variational inference (VI) (Sun et al., 2019; Rudner et al., 2022b). VI is an appealing method because of its successful application to weight-space BNNs, its flexibility in terms of approximate posterior parameterization, and its scalability to large datasets and models (Hoffman et al., 2013; Blundell et al., 2015; Tomczak et al., 2020). Unfortunately, two intractabilities prevent the direct application of function-space VI to BNNs: (i) the distribution of functions generated by the BNN has no explicit density and (ii) computing the Kullbach-Leibler (KL) divergence term in the VI objective (ELBO) involves a supremum over infinitely many subsets. Prior work has proposed to address problem (i) by either using implicit score function estimators (Sun et al., 2019) or linearizing the BNN (Rudner et al., 2022b); and problem (ii) by replacing the supremum by an expectation (Sun et al., 2019), or by estimating it from samples (Rudner et al., 2022b).

However, the problem is actually more severe. Not only is the KL divergence intractable, it is infinite in many cases of interest (Burt et al., 2020), such as when the prior is a non-degenerate GP or a BNN

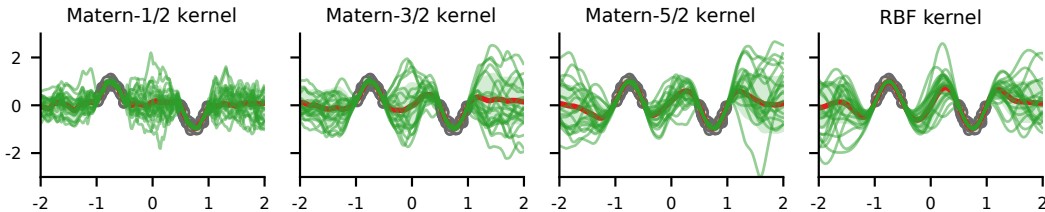

Figure 1: Inference on synthetic data (gray circles) with Gaussian process priors of increasing smoothness from left to right. Our method effectively adapts to the characteristics of each prior.

with a different architecture. Thus, for these situations and actually many more, the KL divergence cannot even be approximated. As a consequence, more recent work abandons the BNN approach and instead uses deterministic neural networks to parameterize basis functions (Ma & Hernández-Lobato, 2021) or a GP mean (Wild et al., 2022). The only prior work (Rudner et al., 2022b) on function-space VI in BNNs that overcomes the issue pointed out by Burt et al. (2020) does so by deliberately limiting itself to cases where the KL divergence is known to be finite (by defining the prior as the pushforward of a weight-space distribution). Therefore, the method by Rudner et al. (2022b) suffers from the same issues regarding prior specification as any other weight-space inference method for BNNs.

In this paper, we follow the argument by Burt et al. (2020) that VI does not provide a valid objective for inference in BNNs with function-space priors, and we propose to apply the framework of generalized variational inference (Knoblauch et al., 2019). We present a simple method for function space inference with GP priors that builds on the linearized BNN setup from Rudner et al. (2022b). Our proposal exploits that, within this linearized framework, the variational posterior induces a Gaussian measure, and so the so-called *regularized* KL divergence (Quang, 2019) is well defined and finite. The regularized KL divergence generalizes the conventional KL divergence, and it allows to use any GP prior for which there exists an equivalent Gaussian measure on the chosen function space. While the regularized KL divergence is still intractable, it can be consistently estimated from samples with a known error bound. We find that our method effectively incorporates the beliefs defined by GP priors into the inference process (see Figure 1). Our contributions are summarized below:

1. We present a new, well-defined objective for function space inference in the linearized BNN with GP priors. The proposed method is based on generalized VI with the so-called *regularized* KL divergence (Quang, 2019).

2. We analyze our method empirically on synthetic and real-world data sets, and find that it approximates the true GP posterior and provides competitive uncertainty estimates for regression and out-of-distribution detection compared to baselines with function and weight-space priors.

The paper is structured as follows. Section 2 introduces function space variational inference in BNNs and discusses its pathologies. Section 3 presents the regularized KL divergence and our proposed method for generalized function-space VI (GFSVI) in BNNs. Section 4 presents experimental results on synthetic and real-world data. We discuss related work in Section 5 and conclude in Section 6.

## 2 FUNDAMENTAL ISSUES WITH FUNCTION SPACE VARIATIONAL INFERENCE

In this section, we briefly state the problem setup of function-space variational inference (Subsection 2.1) and then highlight three important issues (Subsection 2.2). Section 3 proposes a solution.

### 2.1 PROBLEM SETUP AND NOTATION

We consider a neural network $f(\cdot\,; w)$ with weights $w \in \mathbb{R}^p$, and a data set $\mathcal{D} = \{(x_i, y_i)\}_{i=1}^N$ with features $x_i \in \mathcal{X} \subset \mathbb{R}^d$ and associated values $y_i \in \mathbb{R}$. Bayesian Neural Networks (BNNs) require the specification of a likelihood function $p(\mathcal{D} \,|\, w) = \prod_{i=1}^N p(y_i \,|\, f(x_i; w))$ and (traditionally) a prior $p(w)$ on the model weights to compute the posterior distribution $p(w \,|\, \mathcal{D}) \propto p(\mathcal{D} \,|\, w)\, p(w)$. The method proposed in this paper builds on variational inference (VI), which approximates $p(w \,|\, \mathcal{D})$

with a variational distribution $q_\phi(w)$ chosen to maximize the evidence lower bound (ELBO),

$$\mathcal{L}(\phi) := \mathbb{E}_{q_\phi(w)}[\log p(\mathcal{D} \,|\, w)] - D_{\mathrm{KL}}(q_\phi \,\|\, p) \quad \text{with} \quad D_{\mathrm{KL}}(q_\phi \,\|\, p) := \mathbb{E}_{q_\phi(w)}\left[\log \frac{q_\phi(w)}{p(w)}\right]. \quad (1)$$

Here, $D_{\mathrm{KL}}$ is the Kullback-Leibler (KL) divergence. At test time, we approximate the predictive distribution for given features $x^*$ as $p(y^* \,|\, x^*) = \mathbb{E}_{p(w\,|\,\mathcal{D})}\big[p(y^* \,|\, f(x^*; w))\big] \approx \mathbb{E}_{q_\phi(w)}\big[p(y^* \,|\, f(x^*; w))\big]$.

**Function space variational inference.** Since weights of neural networks are not interpretable, we abandon the weight space prior $p(w)$ and instead pose a prior $\mathbb{P}$ directly on the function $f(\,\cdot\,; w)$, which we denote simply as $f$ when there is no ambiguity. Here, the double-lined notation $\mathbb{P}$ denotes a probability measure that does not admit a density since the function space is infinite-dimensional. For the approximate posterior, we still use a variational distribution $q_\phi(w)$ to estimate the expected log-likelihood (first term on the right-hand side of Eq. 1). However for the KL-term, we use the pushforward of $q_\phi(w)$ along the mapping $w \mapsto f(\,\cdot\,; w)$, which defines the variational measure $\mathbb{Q}_\phi$, resulting in the ELBO in function space,

$$\mathcal{L}(\phi) := \mathbb{E}_{q_\phi(w)}[\log p(\mathcal{D} \,|\, w)] - D_{\mathrm{KL}}(\mathbb{Q}_\phi \,\|\, \mathbb{P}) \quad \text{with} \quad D_{\mathrm{KL}}(\mathbb{Q}_\phi \,\|\, \mathbb{P}) = \int \log\left(\frac{d\mathbb{Q}_\phi}{d\mathbb{P}}(f)\right) d\mathbb{Q}_\phi \quad (2)$$

Here, the Raydon-Nikodym derivative $d\mathbb{Q}_\phi/d\mathbb{P}$ generalizes the density ratio from Eq. 1. Like Eq. 1, the ELBO in Eq. 2 is a lower bound on the evidence (Burt et al., 2020). In fact, if $\mathbb{P}$ is the push-forward of $p(w)$ then Eq. 2 is a tighter bound than Eq. 1 by the data processing inequality, $D_{\mathrm{KL}}(\mathbb{Q}_\phi \,\|\, \mathbb{P}) \leq D_{\mathrm{KL}}(q_\phi \,\|\, p)$. However, we motivated function space VI to avoid weight space priors, and in this case the bound in Eq. 2 can be looser. In the next subsection, we see that the bound becomes infinitely loose in practice, and we therefore propose a different objective in Section 3.

## 2.2 THREE ISSUES WITH FUNCTION SPACE VARIATIONAL INFERENCE

We highlight two issues with function space VI that were addressed in the literature (issues (i) and (ii) below), and a third issue which we argue is more severe, and which is the focus of this paper.

**Issue (i): $\mathbb{Q}_\phi$ is intractable.** We cannot express the pushforward measure $\mathbb{Q}_\phi$ of $q_\phi(w)$ in a closed form because the neural network is nonlinear. To mitigate this issue, previous work has proposed using implicit score function estimators (Sun et al., 2019) or a linearized BNN (Rudner et al., 2022a;b). Our proposal in Section 3 follows the linearized BNN approach as it only minimally modifies the BNN, preserving most of its inductive bias (Maddox et al., 2021) while considerably simplifying the problem by turning the pushforward of $q_\phi(w)$ into a GP. More specifically, we consider a Gaussian variational distribution $q_\phi(w) = \mathcal{N}(m, S)$ with parameters $\phi = \{m, S\}$, and we define a linearized neural network $f_L$ by linearizing $f$ as a function of the weights around $w = m$,

$$f_L(x; w) := f(x; m) + J(x, m)(w - m) \qquad \text{where} \qquad J(x, m) = \nabla_w f(x; w)|_{w=m}. \quad (3)$$

Thus, for $w \sim q_\phi(w)$ we have for all $x$ that $f_L(x; w) \sim \mathcal{N}\big(f(x; m), J(x, m)SJ(x, m)^\top\big)$ and that the function $f_L(\,\cdot\,, w)$ is a degenerate GP (since $\mathrm{rank}(J(\mathbf{x}, m)SJ(\mathbf{x}, m)^\top) < \infty$ for any $\mathbf{x} \in \mathcal{X}^M$, $M \in \mathbb{N}$),

$$f_L \sim \mathcal{GP}\big(f(\,\cdot\,; m), J(\,\cdot\,, m)SJ(\,\cdot\,, m)^\top\big) \qquad (\text{for } w \sim q_\phi(w)). \quad (4)$$

**Issue (ii): computing $D_{\mathrm{KL}}(\mathbb{Q}_\phi \,\|\, \mathbb{P})$ is intractable.** It is not obvious how to practically evaluate or estimate a KL divergence between two measures in function space. Sun et al. (2019) showed that it can be expressed as a supremum of KL divergences between finite-dimensional densities,

$$D_{\mathrm{KL}}(\mathbb{Q}_\phi \,\|\, \mathbb{P}) = \sup_{M \in \mathbb{N}, \mathbf{x} \in \mathcal{X}^M} D_{\mathrm{KL}}(q_\phi(f(\mathbf{x})) \,\|\, p(f(\mathbf{x}))). \quad (5)$$

Here, $\mathbf{x} = (x^{(i)})_{i=1}^M \in \mathcal{X}^M$ is a set of $M$ points in feature space $\mathcal{X}$, and $q_\phi(f(\mathbf{x}))$ and $p(f(\mathbf{x}))$ are densities of the marginals of $\mathbb{Q}_\phi$ and $\mathbb{P}$ on $(f(x^{(i)}))_{i=1}^M$, respectively (in our case, $q_\phi(f(\mathbf{x}))$ and $p(f(\mathbf{x}))$ are multivariate normal distributions since $\mathbb{Q}_\phi$ and $\mathbb{P}$ are induced by GPs). To obtain a tractable approximation of the supremum over infinitely many sets in Eq. 5, Sun et al. (2019) replace it by an expectation and Rudner et al. (2022b) estimate it from samples.

**Issue (iii):** $D_{\mathrm{KL}}(\mathbb{Q}_\phi \,\|\, \mathbb{P})$ **is infinite in practically relevant cases.** Burt et al. (2020) point out an even more severe issue of function-space VI in BNNs: $D_{\mathrm{KL}}(\mathbb{Q}_\phi \,\|\, \mathbb{P})$ is in fact infinite in many relevant cases, in particular for non-degenerate GP-priors with most parametric models including BNNs, making approximation attempts in these cases futile. The proof in Burt et al. (2020) is somewhat involved, but the fundamental reason for $D_{\mathrm{KL}}(\mathbb{Q}_\phi \,\|\, \mathbb{P}) = \infty$ is that $\mathbb{Q}_\phi$ has support only on a finite-dimensional submanifold of the infinite dimensional function space, while a non-degenerate GP prior $\mathbb{P}$ has support on the entire function space. That such a dimensionality mismatch can render the KL divergence infinite can already be seen in a simple finite-dimensional example: consider the KL-divergence between two multivariate Gaussians in $\mathbb{R}^n$ for some $n \geq 2$, one of which has support on the entire $\mathbb{R}^n$ (i.e., its covariance matrix $\Sigma_1$ has full rank) while the other one has support only on a proper subspace of $\mathbb{R}^n$ (i.e., its covariance matrix $\Sigma_2$ is singular). The KL divergence between two multivariate Gaussians can be calculated in a closed form expression (see Eq 8) that contains the term $\log(\det \Sigma_1 / \det \Sigma_2)$, which is infinite for singular $\Sigma_2$.

We find that the fact that $D_{\mathrm{KL}}(\mathbb{Q}_\phi \,\|\, \mathbb{P}) = \infty$ has severe practical consequences even when the KL divergence is only estimated from samples. It naturally explains the stability issues discussed in Appendix D.1 of Sun et al. (2019). We discuss differences between the authors' solution and ours at the end of Section 3.3. Surprisingly, similar complications arise even in the setup by Rudner et al. (2022b), which performs VI in function space with the pushforward of a weight-space prior. While the KL divergence in this setup is technically finite because prior and variational posterior have the same support, numerical errors lead to mismatching supports and thus to stability issues even here.

In summary, the ELBO for VI in BNNs is not a well-defined objective for a large class of interesting function-space priors. In the next section, we propose a solution by regularizing the KL divergence.

## 3 GENERALIZED FUNCTION SPACE VI WITH REGULARIZED KL DIVERGENCE

At the end of Section 2.2, we pointed out that the KL divergence in the function-space ELBO (Eq. 2) is often infinite (Burt et al., 2020). Rudner et al. (2022b) addressed this problem by restricting the prior to be the pushforward of a weight-space distribution, which guarantees that the KL divergence is finite but effectively resorts to weight-space priors. In this section, we propose an orthogonal approach in which we take the infinite KL divergence as an indication that variational inference is too restrictive if one wants to use genuine function-space priors. We instead consider generalized variational inference (Knoblauch et al., 2019), which reinterprets the ELBO in Eq. 2 as a regularized expected log-likelihood and explores alternative divergences for the regularizer. This is motivated by the observation that the main assumptions underlying Bayesian inference are only loosely satisfied in BNNs. Specifically, we propose to use the regularized KL divergence proposed by Quang (2019) together with the linearized BNN from Rudner et al. (2022b) (see Eqs. 3 and 4). Below, we first provide background on Gaussian measures (Subsection 3.1), which we use to introduce the regularized KL divergence (Subsection 3.2), and we then present the proposed inference method (Subsection 3.3).

### 3.1 GAUSSIAN MEASURES

The regularized KL divergence defined in Section 3.2 is defined in terms of Gaussian measures, and thus we need to verify that the GP variational posterior induced by the linearized BNN (Eq 4) has an associated Gaussian measure. We consider the Hilbert space $L^2(\mathcal{X}, \rho)$ of square-integrable functions with respect to a probability measure $\rho$ on a compact set $\mathcal{X} \subseteq \mathbb{R}^d$, with inner product $\langle f, g \rangle = \int_\mathcal{X} f(x)g(x)d\rho(x)$. This is not a restrictive assumption as we can typically bound the region in feature space that contains the data and any points where we might want to evaluate the BNN.

**Definition 3.1** (Gaussian measure, Kerrigan et al. (2023), Definition 1). Let $(\Omega, \mathcal{B}, \mathbb{P})$ be a probability space. A measurable function $F : \Omega \mapsto L^2(\mathcal{X}, \rho)$ is called a Gaussian random element (GRE) if for any $g \in L^2(\mathcal{X}, \rho)$ the random variable $\langle g, F \rangle$ has a Gaussian distribution on $\mathbb{R}$. For every GRE $F$, there exists a unique mean element $m \in L^2(\mathcal{X}, \rho)$ and a finite trace linear covariance operator $C : L^2(\mathcal{X}, \rho) \mapsto L^2(\mathcal{X}, \rho)$ such that $\langle g, F \rangle \sim \mathcal{N}(\langle g, m \rangle, \langle Cg, g \rangle)$ for all $g \in L^2(\mathcal{X}, \rho)$. The pushforward of $\mathbb{P}$ along $F$ denoted $\mathbb{P}^F = F_\# \mathbb{P}$ is a Gaussian (probability) measure on $L^2(\mathcal{X}, \rho)$.

Gaussian measures generalizes Gaussian distributions to infinite-dimensional spaces where measures do not have associated densities. Following Wild et al. (2022), we notate the *Gaussian* measure obtained from the GRE $F$ with mean element $m$ and covariance operator $C$ as $\mathbb{P}^F = \mathcal{N}(m, C)$.

**Connection to GPs.** A GP $f \sim \mathcal{GP}(\mu, K)$ has an associated Gaussian measures in $L^2(\mathcal{X}, \rho)$ if its mean function satisfies $\mu \in L^2(\mathcal{X}, \rho)$ and its covariance function $K$ is trace-class i.e. $\int_{\mathcal{X}} K(x, x) d\rho(x) < \infty$ (Wild et al. (2022) Theorem 1). The GP variational posterior induced by the linearized BNN (Eq 4) satisfies both properties as neural networks are well-behaved functions on the compact subset $\mathcal{X}$, and thus induces a Gaussian measure $\mathbb{Q}_\phi^F \sim \mathcal{N}(m_Q, C_Q)$. It turn out that the infinite KL divergence discussed in Section 2.2 is easier to prove for Gaussian measures, and we provide the proof in Appendix A.1.1 for the interested reader.

## 3.2 THE REGULARIZED KL DIVERGENCE.

We now discuss the regularized KL divergence (Quang, 2019), which we will use in Section 3.3.

**Definition 3.2** (Regularized KL divergence, Quang (2022) Definition 5)**.** Let $\mu_1, \mu_2 \in L^2(\mathcal{X}, \rho)$ and $C_1, C_2$ be bounded, self-adjoint, positive and trace-class linear operators on $L^2(\mathcal{X}, \rho)$. Let $\gamma \in \mathbb{R}$, $\gamma > 0$ be fixed. The regularized KL divergence is defined as

$$D_{\mathrm{KL}}^\gamma(\mathcal{N}(\mu_1, C_1) \,\|\, \mathcal{N}(\mu_2, C_2)) = \frac{1}{2} \langle \mu_1 - \mu_2, (C_2 + \gamma\mathbb{I})^{-1}(\mu_1 - \mu_2) \rangle$$
$$+ \frac{1}{2} \mathrm{Tr}_X \left[ (C_2 + \gamma\mathbb{I})^{-1}(C_1 + \gamma\mathbb{I}) - \mathbb{I} \right] - \frac{1}{2} \log \det{}_X \left[ (C_2 + \gamma\mathbb{I})^{-1}(C_1 + \gamma\mathbb{I}) \right] \quad (6)$$

For any $\gamma > 0$, the regularized KL divergence is well-defined and finite, even if the Gaussian measures are singular (Quang, 2019), and it converges to the conventional KL divergence in the limit of $\gamma \to 0$ if the latter is well-defined (Quang, 2022, Theorem 6). Furthermore, if Gaussian measures $\nu_1$ and $\nu_2$ are induced by GPs $f_i \sim \mathcal{GP}(\mu_i, K_i)$ for $i = 1, 2$, then $D_{\mathrm{KL}}^\gamma(\nu_1 \,\|\, \nu_2)$ can be consistently estimated (Quang, 2022) from a finite number $M$ of samples using the estimator

$$\hat{D}_{\mathrm{KL}}^\gamma(\nu_1 \,\|\, \nu_2) := D_{\mathrm{KL}}\big(\mathcal{N}(m_1, \Sigma_1 + \gamma M \,\mathbb{I}_M) \,\|\, \mathcal{N}(m_2, \Sigma_2 + \gamma M \,\mathbb{I}_M)\big) \quad (7)$$

where $m_i := \mu_i(\mathbf{x})$ and $\Sigma_i := K_i(\mathbf{x}, \mathbf{x})$ are the mean vector and covariance matrix obtained by evaluating $\mu_i$ and $K_i$, respectively, at measurement points $\mathbf{x} = [x^{(1)}, \ldots, x^{(M)}]$, $x^{(1)}, \ldots, x^{(M)} \stackrel{\text{i.i.d}}{\sim} \rho(x)$. The KL-divergence on the right-hand side of Eq. 7 is between multivariate Gaussian distributions $p_1 = \mathcal{N}(m_1, \Sigma_1^{(\gamma)})$ and $p_2 = \mathcal{N}(m_2, \Sigma_2^{(\gamma)})$ with $\Sigma_i^{(\gamma)} = \Sigma_i + \gamma M \,\mathbb{I}_M$, evaluated in closed form as

$$D_{\mathrm{KL}}(p_1 \,\|\, p_2) = \frac{1}{2} \left[ \mathrm{Tr} \left[ (\Sigma_2^{(\gamma)})^{-1} \Sigma_1^{(\gamma)} \right] - M + (m_1 - m_2)^\top (\Sigma_2^{(\gamma)})^{-1} (m_1 - m_2) + \log \frac{\det \Sigma_1^{(\gamma)}}{\det \Sigma_2^{(\gamma)}} \right]$$
$$(8)$$

Quang (2022) shows that the absolute error of the estimator is bounded by $\mathcal{O}(1/M)$ with high probability with constants depending on $\gamma$ and properties of the GP mean and covariance functions. We provide the exact bound in Appendix A.1.2.

## 3.3 GENERALIZED FUNCTION SPACE VARIATIONAL INFERENCE

We now present generalized function space variational inference (GFSVI), a simple and practical algorithm to perform generalized variational inference in function space with GP priors. The method starts from the function space ELBO (Eq. 2) with the linearized BNN approximation from Eq. 4 and replaces the KL divergence with the regularized KL divergence $D_{\mathrm{KL}}^\gamma$ discussed in Section 3.2. Assuming a Gaussian likelihood $p(\mathcal{D} \,|\, w) = \prod_{i=1}^N \mathcal{N}\big(f(x_i; w), \sigma_y^2\big)$ and a Gaussian variational distribution $q_\phi(w) = \mathcal{N}(w \,|\, m, S)$, we thus obtain the objective function

$$\mathcal{L}(\phi) = \sum_{i=1}^N \mathbb{E}_{q_\phi(w)} \big[ \log \mathcal{N}\big(y_i \,|\, f_L(x_i; w), \sigma_y^2\big) \big] - D_{\mathrm{KL}}^\gamma\big(\mathbb{Q}_\phi^F \,\|\, \mathbb{P}^F\big) \quad (9)$$

where $\mathbb{Q}_\phi^F$ and $\mathbb{P}^F$ are, respectively, the Gaussian measures corresponding to the GP variational posterior (Eq. 4), and a GP prior $f_L \sim \mathcal{GP}(\mu, K)$ that satisfies the conditions discussed in Section 3.1.

We maximize the objective in Eq. 9 over the mean $m$ and covariance matrix $S$ of the Gaussian variational distribution $q_\phi(w)$, and over the likelihood scale parameter $\sigma_y$, see Algorithm 1. Note that, unlike Rudner et al. (2022b), which uses the standard BNN and Monte-Carlo integration to estimate

---

**Algorithm 1:** Generalized function space variational inference (GFSVI)

---

**Input:** BNN $f$, GP prior $\mathcal{GP}(\mu, K)$, measurement point distribution $\rho(x)$, $\gamma > 0$, batch size $B$.

**Data:** $\mathcal{D} = \{(x_i, y_i)\}_{i=1}^N$.

**for** each minibatch $(x_\mathcal{B}, y_\mathcal{B}) \sim \mathcal{D}$ **do**

  Draw measurement set $\mathbf{x} = [x^{(1)}, \ldots, x^{(M)}]$, $x^{(1)}, \ldots, x^{(M)} \overset{\text{i.i.d}}{\sim} \rho(x)$

  Calculate $\mathcal{L}(\phi) = \frac{N}{B} \mathbb{E}_{q_\phi(w)}\left[\log \mathcal{N}\left(y_\mathcal{B} \mid f_L(x_\mathcal{B}; w), \sigma_y^2\right)\right] - \hat{D}_{\text{KL}}^\gamma\left(\mathbb{Q}_\phi^F \,\|\, \mathbb{P}^F\right)$   *// Eqs. 10 & 11*

  Update $\phi$ using a gradient step in the direction $\nabla_\phi \mathcal{L}(\phi)$

**end**

---

the expected log-likelihood (first term in Eq 9), we also consider the linearized BNN in this term, as this has proven more stable during training, and it provides an analytical expression

$$\mathbb{E}_{q_\phi(w)}\left[\log \mathcal{N}\left(y_i \mid f_L(x_i; w), \sigma_y^2\right)\right] = -\frac{1}{2}\log\left(2\pi\sigma_y^2\right) - \frac{(y_i - f(x_i; m))^2 + J(x_i; m)SJ(x_i; m)^\top}{2\sigma_y^2}. \tag{10}$$

We further use the consistent estimator for the regularized KL divergence

$$\hat{D}_{\text{KL}}^\gamma\left(\mathbb{Q}_\phi^F \,\|\, \mathbb{P}^F\right) = D_{\text{KL}}\left(\mathcal{N}\left(f(\mathbf{x}), J(\mathbf{x})SJ(\mathbf{x})^\top + \gamma M \mathbb{I}_M\right) \,\|\, \mathcal{N}\left(\mu(\mathbf{x}), K(\mathbf{x}, \mathbf{x}) + \gamma M \mathbb{I}_M\right)\right) \tag{11}$$

with measurement points $\mathbf{x} = [x^{(1)}, \ldots, x^{(M)}]$, $x^{(1)}, \ldots, x^{(M)} \overset{\text{i.i.d}}{\sim} \rho(x)$ sampled from a probability measure on $\mathcal{X}$, and $f$ and $J$ are evaluated at the posterior mean $m$ as in Eq 4.

**Technical details ($\gamma$ and $\rho$).** The $\gamma$ parameter both controls the magnitude of the regularized KL divergence (see Figure 12 in Appendix) and acts as jitter. We recommend choosing $\gamma$ large enough to avoid numerical errors while remaining sufficiently small to provide strong regularization (see Figure 10 in Appendix). Furthermore, the probability measure $\rho$ defined with $L^2(\mathcal{X}, \rho)$ has to assign non-zero probability to any open set of $\mathcal{X}$ to regularize the BNN on all of its support. Following Rudner et al. (2022b), we sample measurement points from a uniform distribution over $\mathcal{X}$.

**Differences to prior work.** Both TFSVI (Rudner et al., 2022b) and FVI (Sun et al., 2019) solve stability issues by introducing jitter/noise, which has a similar effect as the regularization in Equation (6). However, as mentioned at the end of Section 2.2, TFSVI only introduces jitter to overcome numerical issues and is fundamentally restricted to prior specification in weight space. FVI does not linearize the BNN, and therefore does not have access to an explicit variational measure in function space. This severely complicates the estimation of (gradients of) the KL divergence in FVI, and the authors resort to implicit score function estimators, which make FVI difficult to use in practice (Ma & Hernández-Lobato, 2021). Our proposed GFSVI does not suffer from these complications as the variational posterior is an explicit (Gaussian) measure. This allows us to estimate the (regularized) KL divergence without sampling noise or having to use implicit score function estimators.

## 4 EXPERIMENTS

We evaluate our generalized function space variational inference (GFSVI) method qualitatively on synthetic data (Subsection 4.1) and quantitatively on real-world data (Subsection 4.2). We find that GFSVI approximates the exact Gaussian process (GP) posterior more faithfully than all our baselines, and that it performs competitively on regression and out-of-distribution detection tasks (like in Sun et al. (2019), we do not consider classification tasks as prior specification in function space does not provide much advantage here). We also discuss the influence of the BNN's inductive biases.

**Baselines.** We compare the proposed GFSVI method to two weight-space inference methods (mean-field variational inference (Blundell et al., 2015) and linearized Laplace (Immer et al., 2021)) and to two function-space inference methods (FVI (Sun et al., 2019) and TFVSI (Rudner et al., 2022b)), where the latter performs inference in function space but with the pushforward of a weight space prior). All BNNs have the same architecture and fully-factorized Gaussian approximate posterior. We include results for Gaussian Process (GP) (Williams & Rasmussen, 2006) when the size of the

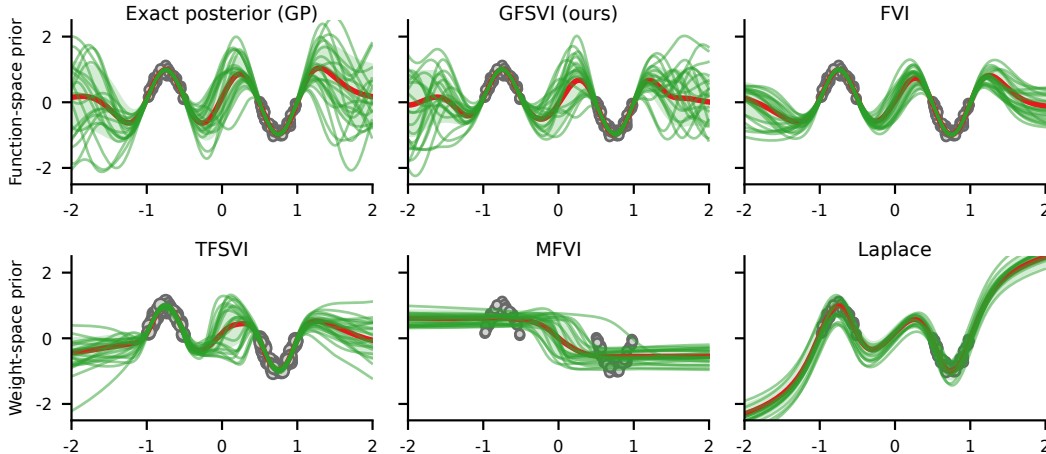

Figure 2: Inference on synthetic data (gray circles) using an RBF prior for function space methods GFSVI and FVI. The proposed GFSVI provides the best approximation of the exact GP posterior.

data set allows it, and for a sparse GP (Hensman et al., 2013). We consider the GP (and sparse GP) as gold standards as they represent the exact (or near exact) posterior for models with GP priors.

## 4.1 QUALITATIVE RESULTS ON SYNTHETIC DATA

We consider synthetic data with a one-dimensional feature space $\mathcal{X} = \mathbb{R}$, where the values $y_i$ are randomly sampled around $\sin(2\pi x_i)$ (circles in Figures 1-3, see Appendix A.2.1). The red lines in Figures 1-3 show inferred mean functions, and green lines are sampled functions from (approximate) posteriors. Figure 2 compares GFSVI with an RBF GP prior to all baselines and to the exact posterior. We find that GFSVI visually matches the true posterior best. This is further supported by Figure 4 in the Appendix, which uses a Matern-1/2 prior. Figure 1 in the Introduction and Figure 6 in the Appendix show that GFSVI notably adapts to varying prior assumptions (varying smoothness assumptions in Figure 1 and varying length-scale in Figure 6). In addition, Figures 7 and 5 in the Appendix show that GFSVI provides strong regularization when the data generative process is noisy, and that it can be trained with fewer measurement points $M$ than FVI without significant degradation.

**Inductive biases.** Figure 3 compares our model to the exact posterior across two different priors and three model architectures (details in Appendix A.2.1). We find that, when using piece-wise linear activations (ReLU), small models are prone to underfitting for smooth priors (RBF), and to collapsing uncertainty for rough priors (Matern-1/2). By contrast, when using smooth activations (Tanh), smaller models suffice, and they are compatible with most standard GP priors (the results shown in Figure 3 extend to RBF, Matern family, and Rational Quadratic in our experiments). We also analyzed how the number $M$ of measurement points affects performance. Figures 8 and 9 in the Appendix show that capturing the properties of rough GP priors and estimating $\tilde{D}_{\mathrm{KL}}^{\gamma}$ with these priors requires larger $M$.

## 4.2 QUANTITATIVE RESULTS ON REAL-WORLD DATA

We evaluate GFSVI on data sets from the UCI repository (Dua & Graff, 2017) described in Table 4 in the Appendix. We perform 5-fold cross validation and report mean and standard deviation of the scores across the test folds. When reporting results, we bold the highest score, as well as any score if its error bar and the highest score's error bar overlap, considering the difference to not be statistically significant. We also report the mean rank of the methods across datasets.

**Regression.** We evaluate the predictive performance of our model and report the average expected log-likelihood and average mean square error (MSE) on the test folds. Additional details are provided in Appendix A.2.2. We find that GFSVI performs competitively on regression tasks compared to baselines and obtains the best mean rank, matching the top performing methods on nearly all datasets (see Table 1 and Table 5 in Appendix). In particular, we find that using GP priors in the linearized

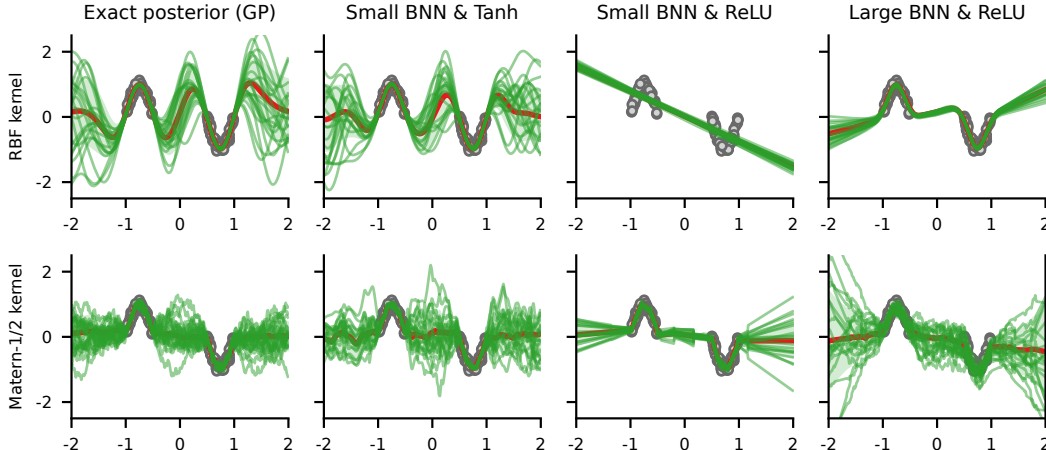

Figure 3: Our method requires that the Bayesian neural network (BNN) and Gaussian process (GP) prior share similar inductive biases to provide an accurate approximation to the exact posterior.

Table 1: Test log-likelihood (higher is better) of evaluated methods on regression datasets. GFSVI performs competitively on regression tasks compared to baselines and obtains the best mean rank.

| Dataset | Function space priors | | Weight space priors | | | Gold standards | |
|---|---|---|---|---|---|---|---|
| | GFSVI (ours) | FVI | TFSVI | MFVI | Laplace | Sparse GP | GP |
| Boston | **-0.733 ± 0.321** | **-0.571 ± 0.253** | -1.416 ± 0.103 | -1.308 ± 0.117 | **-0.812 ± 0.458** | -0.884 ± 0.408 | -1.594 ± 1.243 |
| Concrete | **-0.457 ± 0.092** | **-0.390 ± 0.038** | -0.983 ± 0.027 | -1.353 ± 0.041 | -0.715 ± 0.057 | -0.966 ± 0.057 | -2.099 ± 0.942 |
| Energy | **1.319 ± 0.116** | **1.377 ± 0.094** | 0.797 ± 0.218 | -0.926 ± 0.440 | **1.304 ± 0.096** | -0.206 ± 0.060 | -0.205 ± 0.050 |
| Kin8nm | **-0.136 ± 0.030** | -0.141 ± 0.051 | **-0.182 ± 0.025** | -0.641 ± 0.503 | -0.285 ± 0.031 | -0.443 ± 0.031 | - |
| Naval | **3.637 ± 0.296** | 2.165 ± 0.433 | 2.758 ± 0.097 | 1.034 ± 0.358 | **3.404 ± 0.187** | 4.951 ± 0.032 | - |
| Power | **0.044 ± 0.024** | 0.031 ± 0.048 | 0.007 ± 0.027 | -0.003 ± 0.034 | -0.002 ± 0.043 | -0.100 ± 0.022 | - |
| Protein | -1.036 ± 0.012 | -1.045 ± 0.011 | **-1.010 ± 0.009** | -1.112 ± 0.016 | -1.037 ± 0.015 | -1.035 ± 0.005 | - |
| Wine | **-1.289 ± 0.090** | **-1.215 ± 0.016** | -2.138 ± 0.495 | -1.248 ± 0.040 | **-1.249 ± 0.055** | -1.240 ± 0.083 | -1.219 ± 0.079 |
| Yacht | **1.058 ± 0.180** | 0.545 ± 1.643 | -1.187 ± 0.144 | -1.638 ± 0.066 | **0.680 ± 0.382** | -0.979 ± 0.206 | -0.914 ± 0.100 |
| Wave | 5.521 ± 0.081 | 6.612 ± 0.018 | 5.148 ± 0.261 | **6.883 ± 0.019** | 4.658 ± 0.060 | 4.909 ± 0.002 | - |
| Denmark | **-0.487 ± 0.028** | -0.801 ± 0.012 | **-0.513 ± 0.028** | -0.675 ± 0.015 | -0.600 ± 0.018 | -0.768 ± 0.002 | - |
| Mean rank | 1.273 | 1.636 | 1.909 | 2.364 | 1.727 | - | - |

BNN with GFSVI yields improvements over the weight space priors used in TFSVI and that GFSVI performs slightly better than FVI.

Table 2: Average point-wise Wasserstein-2 distance (lower is better) between exact and approximate posterior of reported methods. GFSVI (ours) provides a more accurate approximation than FVI.

| Dataset | Boston | Concrete | Energy | Wine | Yacht | Mean rank |
|---|---|---|---|---|---|---|
| GFSVI (ours) | **0.0259 ± 0.0089** | **0.0499 ± 0.0064** | **0.0035 ± 0.0008** | **0.0571 ± 0.0216** | **0.0036 ± 0.0014** | 1.0 |
| FVI | 0.0469 ± 0.0098 | 0.0652 ± 0.0083 | **0.0037 ± 0.0009** | 0.1224 ± 0.0373 | **0.0052 ± 0.0029** | 1.6 |
| GP sparse | 0.0074 ± 0.0050 | 0.0125 ± 0.0036 | 0.0042 ± 0.0007 | 0.0170 ± 0.0079 | 0.0035 ± 0.0017 | - |

**Conditional sample generation.** We further evaluate our inference method by comparing samples drawn from the exact posterior with the approximate posterior obtained with our method (GFSVI). Additional details are provided in Appendix A.2.3. We find that GFSVI approximates the exact posterior more accurately that FVI obtaining a higher mean rank, but worse than the gold standard sparse GP which demonstrates to be most accurate (see Table 2).

**Out-of-distribution detection.** We further evaluate our method by testing if its epistemic uncertainty is predictive of OOD data following the setup from Malinin et al. (2020). We report the accuracy of a single threshold to classify OOD from ID data based on the predictive uncertainty. Additional details are provided in Appendix A.2.4. We find that GFSVI performs competitively on OOD detection either out-performing or matching best performing baselines, and obtains the highest mean rank (see Table 3). In particular, we find that using GP priors with GFSVI rather than weight space priors with TFSVI is beneficial, and that GFSVI tends to improve over FVI.

Table 3: Out-of-distribution accuracy (higher is better) of evaluated methods on regression datasets. GFSVI (ours) performs competitively on OOD detection and obtains the highest mean rank.

| DATASET | FUNCTION SPACE PRIORS | | WEIGHT SPACE PRIORS | | | GOLD STANDARDS | |
|---|---|---|---|---|---|---|---|
| | GFSVI (OURS) | FVI | TFSVI | MFVI | LAPLACE | SPARSE GP | GP |
| BOSTON | **0.893 ± 0.025** | 0.594 ± 0.054 | **0.705 ± 0.239** | 0.521 ± 0.013 | 0.557 ± 0.021 | 0.947 ± 0.024 | 0.952 ± 0.006 |
| CONCRETE | **0.656 ± 0.036** | **0.583 ± 0.050** | 0.511 ± 0.008 | **0.605 ± 0.027** | 0.578 ± 0.033 | 0.776 ± 0.014 | 0.933 ± 0.010 |
| ENERGY | **0.997 ± 0.004** | 0.696 ± 0.037 | **0.997 ± 0.002** | 0.678 ± 0.032 | 0.782 ± 0.044 | 0.998 ± 0.003 | 0.998 ± 0.002 |
| KIN8NM | 0.588 ± 0.016 | **0.604 ± 0.051** | **0.576 ± 0.018** | **0.570 ± 0.021** | **0.606 ± 0.020** | 0.608 ± 0.031 | - |
| NAVAL | **1.000 ± 0.000** | **1.000 ± 0.000** | **1.000 ± 0.000** | 0.919 ± 0.038 | **1.000 ± 0.000** | 1.000 ± 0.000 | - |
| POWER | **0.698 ± 0.014** | **0.663 ± 0.047** | **0.676 ± 0.019** | 0.636 ± 0.044 | 0.654 ± 0.028 | 0.717 ± 0.010 | - |
| PROTEIN | **0.860 ± 0.024** | **0.810 ± 0.050** | **0.841 ± 0.040** | 0.693 ± 0.046 | 0.629 ± 0.030 | 0.967 ± 0.001 | - |
| WINE | **0.665 ± 0.028** | 0.517 ± 0.009 | 0.549 ± 0.033 | 0.542 ± 0.020 | 0.531 ± 0.016 | 0.781 ± 0.031 | 0.787 ± 0.016 |
| YACHT | 0.616 ± 0.067 | **0.604 ± 0.056** | **0.659 ± 0.097** | **0.642 ± 0.079** | **0.612 ± 0.054** | 0.762 ± 0.041 | 0.788 ± 0.026 |
| WAVE | **0.975 ± 0.011** | 0.642 ± 0.008 | 0.835 ± 0.076 | 0.658 ± 0.058 | 0.529 ± 0.010 | 0.513 ± 0.002 | - |
| DENMARK | 0.521 ± 0.013 | **0.612 ± 0.017** | 0.519 ± 0.012 | 0.225 ± 0.007 | 0.529 ± 0.017 | 0.626 ± 0.005 | - |
| MEAN RANK | 1.182 | 1.545 | 1.456 | 1.909 | 1.909 | - | - |

## 5 RELATED WORK

In this section, we review related work on function space variational inference with neural networks, and on approximating functions-space measures with weight-space priors.

**Function-space inference with neural networks.** Prior work on function space VI in BNNs has addressed issues (i) and (ii) mentioned in Section 2.2. Sun et al. (2019) address (i) (intractable variational posterior in function space) by using implicit score function estimators, and (ii) (intractable KL divergence) by replacing the supremum with an expectation. Rudner et al. (2022a;b) address (i) by using a linearized the BNN (Khan et al., 2020; Immer et al., 2021; Maddox et al., 2021), and (ii) by replacing the supremum with a maximum over a finite set of samples. Other work abandons approximating the posterior over neural network weights altogether and instead uses a BNN only to specify a GP prior (Ma et al., 2019), or deterministic neural networks to fit basis functions for Bayesian linear regression (Ma & Hernández-Lobato, 2021) or the mean of a generalized sparse GP with Wasserstein-2 metric (Wild et al., 2022). Our work combines linearized BNNs with generalized variational inference, but we use the regularized KL divergence (Quang, 2019), which naturally generalizes the conventional KL divergence and allows for informative GP priors.

**Approximating function-space measures with weight-space priors in BNNs.** Flam-Shepherd et al. (2017); Tran et al. (2022) minimize a divergence between the BNN's prior predictive and a GP before performing inference on weights, while Wu et al. (2023) directly incorporate the bridging divergence inside the inference objective. Alternatively, Pearce et al. (2020) derive BNN architectures mirroring GPs, and Matsubara et al. (2022) use the Ridgelet transform to design weight-spaces priors approximating a GP in function space. Another line of work considers weight-space priors which regularize in function space by comparing the model's predictions to those of a reference model (Nalisnick et al., 2020) and using an empirical prior (Rudner et al., 2023).

## 6 DISCUSSION

As a solution to the infinite KL divergence problem in function space VI, we proposed to follow the generalized VI framework and to substitute the KL divergence in the ELBO by the regularized KL divergence which is always finite. We presented a simple and well-defined objective for function space inference in the linearized BNN with GP priors based on this divergence. We demonstrated that our method accurately approximates the true GP posterior on synthetic and small real-world data sets, and provides competitive uncertainty estimates for regression and out-of-distribution detection compared to BNN baselines with both function and weight space priors.

Future work should investigate adapting our method to classification, where many applications may benefit from well-calibrated uncertainty estimates (Shamshirband et al., 2021; Krishnapriya & Karuna, 2023). One could also investigate the use of more expressive variational distributions over weights, such as Gaussian with low-rank plus diagonal covariance proposed by Tomczak et al. (2020).

**Reproducibility Statement.** All code and details necessary to reproduce the results in this paper are provided in supplementary materials.

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

# A APPENDIX

## A.1 ADDITIONAL DETAILS ON DIVERGENCES BETWEEN GAUSSIAN MEASURES

### A.1.1 THE KL DIVERGENCE IS INFINITE

In this section, we show that the Kullbach-Liebler (KL) divergence between the Gaussian measures $\mathbb{Q}_\phi^F$ and $\mathbb{P}^F$, respectively induced by the linearized BNN in Eq 4 and by a non-degenerate Gaussian process satisfying conditions given in Section 3.1, is infinite. While this has already been shown by Burt et al. (2020), the proof is easier for Gaussian measures. We first need the Feldman-Hàjek theorem which tells us when the KL divergence between two Gaussian measures is well defined.

**Theorem 1** (Feldman-Hàjek, Quang (2022) Theorem 2, Simpson (2022) Theorem 7). Consider two Gaussian measures $\mu_1 = \mathcal{N}(m_1, C_1)$ and $\mu_2 = \mathcal{N}(m_2, C_2)$ on $L^2(\mathcal{X}, \rho)$. Then $\mu_1$ and $\mu_2$ are called equivalent if and only if the following holds:

1. $m_1 - m_2 \in \text{Im}(C_2^{1/2})$
2. The operator $T$ such that $C_1 = C_2^{1/2}(I - T)C_2^{1/2}$ is Hilbert-Schmidt, that is $T$ has a countable set of eigenvalues $\lambda_i$ that satisfy $\lambda_i < 1$ and $\sum_{i=1}^\infty \lambda_i^2 < \infty$.

otherwise $\mu_1$ and $\mu_2$ are singular. If $\mu_1$ and $\mu_2$ are equivalent, then the Radon-Nikodym dervative exists and $D_{\text{KL}}(\mu_1 \| \mu_2)$ admits an explicit formula. Otherwise, $D_{\text{KL}}(\mu_1 \| \mu_2) = \infty$.

Let us now show that the KL divergence between $\mathbb{Q}_\phi^F$ and $\mathbb{P}^F$ is indeed infinite.

**Proposition 1.** The Gaussian measures $\mathbb{Q}_\phi$ and $\mathbb{P}$ are mutually singular and $D_{\text{KL}}(\mathbb{Q}_\phi^F \| \mathbb{P}^F) = \infty$.

*Proof.* The proof follows from the Feldman-Hàjek theorem (Theorem 1). In our case, $C_Q$ has at most $p$ non-zero eigenvalues as the covariance function of the GP induced by the BNN is degenerate, while $C_P$ has a countably infinite non-zeros eigenvalues (prior is non-degenerate as per assumption). Hence, for the equality in condition (2) to hold, $T$ must have eigenvalue 1 which violates the requirement that $T$ is Hilbert-Schmidt i.e. that its eigenvalues $\{\lambda_i\}_{i=1}^\infty$ satisfy $\lambda_i < 1$ and $\sum_{i=1}^\infty \lambda_i^2 < \infty$. Therefore, $\mathbb{Q}_\phi$ and $\mathbb{P}$ are mutually singular and $D_{\text{KL}}(\mathbb{Q}_\phi \| \mathbb{P}) = \infty$. □

### A.1.2 THE REGULARIZED KL DIVERGENCE

In this section, we provide the bound describing the asymptotic convergence of the regularized KL divergence estimator.

**Theorem 2** (Convergence of estimator, Quang (2022) Theorem 45). Assume the following:

1. Let T be a $\sigma-$ compact metric space, that is $T = \cup_{i=1}^\infty T_i$, where $T_1 \subset T_2 \subset \cdots$ with each $T_i$ being compact.
2. $\rho$ is a non-degenerate Borel probability measure on T, that is $\rho(B) > 0$ for each open set $B \subset T$.
3. $K_1, K_2 : T \times T \to \mathbb{R}$ are continuous, symmetric, positive definite kernels and $\exists \kappa_1 > 0, \kappa_2 > 0$ such that $\int_T K_i(x, x)d\rho(x) \le \kappa_i^2$ for $i = 1, 2$.
4. $\sup_{x \in T} K_i(x, x) \le \kappa_i^2$ for $i = 1, 2$.
5. $\eta_i \sim GP(\mu_i, K_i)$, where $\mu_i \in L^2(T, v)$ for $i = 1, 2$.
6. $\exists B_i > 0$ such that $\|\mu_i\|_\infty \le B_i$ for $i = 1, 2$.

Let $\mathbf{x} = [x^{(1)}, \dots, x^{(M)}], x^{(1)}, \dots, x^{(M)} \overset{\text{i.i.d}}{\sim} \rho(x)$. If Gaussian measures $\mathcal{N}(\mu_i, C_i)$ are induced by GPs $f_i \sim \mathcal{GP}(\mu_i, K_i)$ for $i = 1, 2$, then for any $0 < \delta < 1$, with probability at least $1 - \delta$,

$$|D_{\text{KL}}(\mathcal{N}(\mu_1(\mathbf{x}), K_1(\mathbf{x}, \mathbf{x}) + M\gamma\mathbb{I}_M) \| \mathcal{N}(\mu_2(\mathbf{x}), K_2(\mathbf{x}, \mathbf{x}) + M\gamma\mathbb{I}_M))$$
$$- D_{\text{KL}}^\gamma(\mathcal{N}(\mu_1, C_1) \| \mathcal{N}(\mu_2, C_2))|$$

$$\le \frac{1}{2\gamma}(B_1 + B_2)^2[1 + \kappa_2^2/\gamma]\left(\frac{2\log\frac{48}{\gamma}}{M} + \sqrt{\frac{2\log\frac{48}{\gamma}}{M}}\right)$$

$$+ \frac{1}{2\gamma^2}[\kappa_1^4 + \kappa_2^4 + \kappa_1^2\kappa_2^2(2 + \kappa_2^2/\gamma)]\left(\frac{2\log\frac{12}{\gamma}}{M} + \sqrt{\frac{2\log\frac{12}{\gamma}}{M}}\right) \quad (12)$$

## A.2 Additional experimental details

### A.2.1 Toy experiments on synthetic data

**Generative model.** We consider the following generative model for the toy data

$$y_i = \sin(2\pi x_i) + \epsilon \quad \text{with} \quad \epsilon \sim \mathcal{N}\left(0, \sigma_n^2\right) \tag{13}$$

and draw $x_i \sim \mathcal{U}([-1, -0.5] \cup [0.5, 1])$. When not otherwise specified, we use $\sigma_n = 0.1$. On the plots, the data points are shown as gray circles, inferred mean functions as red lines, and functions sampled from the approximate posterior as green lines.

**Setup details.** In general, we consider two hidden-layer BNNs with 30 neurons per layer and hyperbolic tangent activation (Tanh) functions. Specifically in Figure 3, the small BNN has the same architecture as above while the large BNN has 100 neurons per layer. All the BNN baselines have the same architecture and fully-factorized Gaussian approximate posterior. The prior scale of TFSVI (Rudner et al., 2022b) is set to $\sigma_p = 0.2$ and $\sigma_p = 0.75$ for MFVI (Blundell et al., 2015) and Laplace (Immer et al., 2021). Apart from the cases where the parameters of the GP prior used for GFSVI (our method) and FVI (Sun et al., 2019) are explicitly stated, we consider a constant zero mean function and find the parameters of the covariance function by maximizing the log-marginal likelihood from mini-batches (Chen et al., 2021). Except where otherwise stated, we estimate the functional KL divergences with 500 measurement points and use the regularized KL divergence with $\gamma = 10^{-10}$.

### A.2.2 Regression experiments on real data

**Datasets and pre-processing** We evaluate the predictive performance of our model on regression datasets from the UCI repository (Dua & Graff, 2017) described in Table 4. These datasets are also considered in Wild et al. (2022); Sun et al. (2019) but we include two additional larger ones (wave, denmark). We perform 5-fold cross validation, leave out one fold for testing, consider 10% of the remaining 4 folds as validation data and the rest as training data. We report mean and standard-deviation of the per-sample average expected log-likelihood and per-sample average mean square error on the test fold. We also report the mean rank of the methods across all datasets by assigning rank 1 to the best scoring method as well as any method who's error bars overlap with the highest score's error bars, and recursively apply this procedure to the methods not having yet been assigned a rank. The expected log-likelihood is estimated by Monte Carlo integration when it is not available in closed form (MFVI, TFSVI and FVI) with 100 posterior samples. We preprocess the dataset by encoding categorical features as one-hot vectors and standardizing the features and labels.

**Baseline specification** We compare our GFSVI method to two weight space inference methods (mean-field variational inference (Blundell et al., 2015) and linearized Laplace (Immer et al., 2021)) and two function space inference methods (FVI (Sun et al., 2019) and TFSVI (Rudner et al., 2022b)). While FVI uses GP priors, TFSVI performs inference in function space but with the pushforward to function space of the variational distribution and prior on the weights. We compute the function space (regularized) KL divergence using a set of 500 measurement points sampled from a uniform distribution for GFSVI and TFSVI, and 50 points drawn from a uniform distribution along with 450 samples from the training batch for FVI as specified in Sun et al. (2019). All the BNN baselines have the same architecture and fully-factorized Gaussian approximate posterior. We also provide results with a GP (Williams & Rasmussen, 2006) when the size of the dataset allows it, and a sparse GP (Hensman et al., 2013). As we restrict our comparison to BNNs, we do not consider the GP and sparse GP as baselines but rather as gold-standards. All models have a Gaussian heteroskedastic noise model with a learned scale parameter. All the BNNs are fit using the Adam optimizer (Kingma & Ba, 2017) using a mini-batch size of 2000 samples. We also perform early stopping when the validation loss stops decreasing.

**Model selection** Hyper-parameter optimization is conducted using the Bayesian optimization tool provided by Wandb (Biewald, 2020). BNN parameters are selected to maximize the mean expected log-likelihood of the validation data across the 5 cross-validation folds. We optimize over prior parameters (kernel, prior scale), learning-rate and activation function. The GP prior parameters used with GFSVI and FVI are selected by maximizing the log-marginal likelihood from batches as proposed by (Chen et al., 2021) and done in Sun et al. (2019). The GPs and sparse GPs kernel

Table 4: Regression dataset description

| DATASET | BOSTON | NAVAL | POWER | PROTEIN | YACHT | CONCRETE | ENERGY | KIN8NM | WINE | WAVE | DENMARK |
|---|---|---|---|---|---|---|---|---|---|---|---|
| NUMBER SAMPLES | 506 | 11 934 | 9 568 | 45 730 | 308 | 1 030 | 768 | 8 192 | 1 599 | 288 000 | 434 874 |
| NUMBER FEATURES | 13 | 16 | 4 | 9 | 6 | 8 | 8 | 8 | 11 | 49 | 2 |

parameters and learning-rate are selected to maximize the log-marginal likelihood of the validation data across the 5 cross-validation folds.

**Software** We use the JAX (Bradbury et al., 2018) and DM-Haiku (Hennigan et al., 2020) Python libraries to implement our Bayesian neural networks. MFVI, linearized Laplace and TFSVI were implemented based on the information in the papers, and code for FVI was adapted to the JAX library from the implementation provided by the authors. We further use the GPJAX Python library for experiments involving Gaussian processes (Pinder & Dodd, 2022).

**Hardware** All models were fit using a single NVIDIA RTX 2080Ti GPU with 11GB of memory.

### A.2.3 VARIATIONAL MEASURE EVALUATION

We further evaluate our inference method by comparing the samples drawn from the exact posterior over functions with the approximate posterior obtained with our method (GFSVI). We follow the setup by Wilson et al. (2022) and we compute the average per-sample Wasserstein-2 metric between 1000 samples drawn from a GP posterior with RBF kernel evaluated at the test points, and samples from the approximate posterior of GFSVI, sparse GP and FVI evaluated at the same points and with the same prior. We consider the boston, concrete, energy, wine and yacht datasets for which the exact GP posterior can be computed and use the same preprocessing as for regression (see Appendix A.2.2). We report the mean and standard deviation of the average per-example Wasserstein-2 metric across the 5 folds of cross-validation. The Wasserstein-2 metric is computed using the Python Optimal Transport library (Flamary et al., 2021).

**Baseline specification** FVI and GFSVI have the same two hidden layer neural network architecture with 100 neurons each and hyperbolic tangent activation. These models are fit with the same learning rate and set of context points jointly sampled from a uniform distribution over the feature space and mini-batch of training samples. We use $\gamma = 10^{-15}$ for the regularized KL divergence. We further consider a sparse GP with 100 inducing points.

### A.2.4 OOD DETECTION

We evaluate our method by testing if it's epistemic uncertainty is predictive of out-of-distribution (OOD) data following the setup from Malinin et al. (2020), taking epistemic uncertainty to be the variance of the mean prediction with respect to samples from the posterior. We consider the test data to be in-distribution (ID) data and a subset of the song dataset (Bertin-Mahieux et al., 2011) of equal length and with an equal number of features as out-of-distribution (OOD) data. We first fit a model, then evaluate the extend by which the epistemic uncertainty under the model is predictive of the ID and OOD data using a single threshold obtained by a depth-1 decision tree fit to minimize the classification loss. We use the same preprocessing as for regression as well as the same baselines with the same hyper-parameters (see Appendix A.2.2). We also provide results obtained using a GP and sparse GP as gold standard. We report the mean and standard deviation of the accuracy of the threshold to classify OOD from ID data based on epistemic uncertainty across the 5 folds of cross-validation.

### A.3 ADDITIONAL EXPERIMENTAL RESULTS

### A.3.1 UNCERTAINTY VISUALIZATION

In order to demonstrate the quality of the predictive uncertainty of our model, we present additional plots of functions sampled from the GFSVI comparing these to samples from baselines. We further find that GFSVI provides strong regularization when the data generative process is noisy (see Figure 7)

and is more robust than FVI to situations where ones computational budget constrains the number of measurement points $M$ to be small (Figure 5). In contrast to FVI, GFSVI accurately approximates the exact GP posterior under rough (Matern-1/2) GP priors effectively incorporating prior knowledge defined by the GP prior to the inference process (see Figure 4). Likewise, GFSVI adapts to the variability of the functions specified by the kernel (see Figure 6). We also find that GFSVI requires a larger number of measurement points to capture the behavior of a rougher prior (see Figure 8).

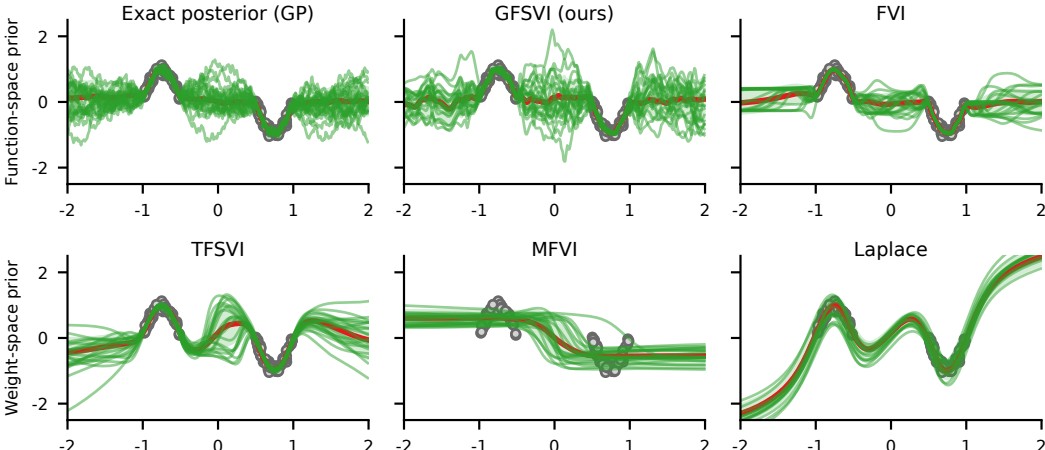

Figure 4: Our method (GFSVI) with a Matern-1/2 Gaussian process (GP) prior accurately approximates the exact GP posterior unlike the function space baseline (FVI). Weight space baselines do not provide a straight-forward mechanism to incorporate prior assumptions regarding the functions generated by BNNs and underestimate the epistemic uncertainty (MFVI, Laplace). The lower row is identical to the one in Figure 2 in the main text and is reproduced here to make comparison easier.

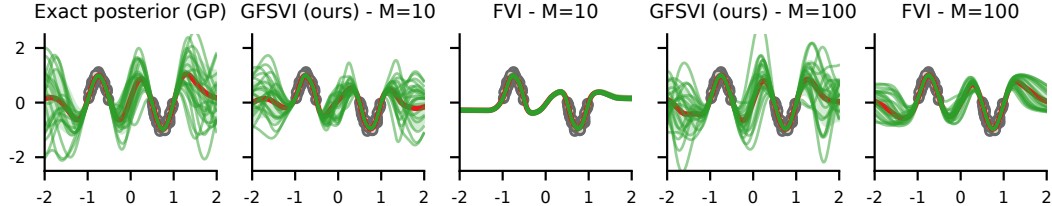

Figure 5: Our method (GFSVI) already provides a reasonable approximation to the exact posterior with small numbers of measurement points (M=10) while function space baseline FVI requires many more (M=100).

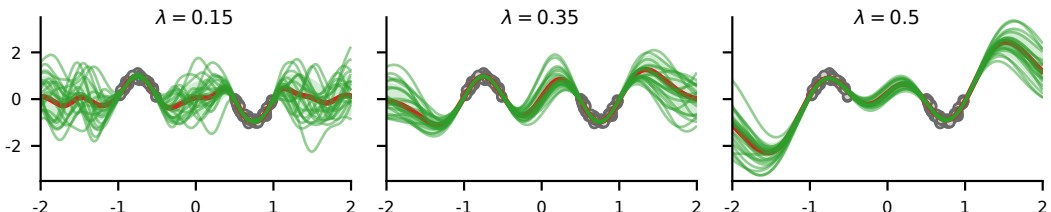

Figure 6: Our method (GFSVI) allows to incorporate prior beliefs in terms of function variability using the characteristic length-scale parameter of the Gaussian process (GP) prior. GFSVI was fit using a GP with RBF covariance function.

### A.3.2 REGRESSION

In this section we present additional regression results reporting the mean square error (MSE) of evaluated methods across the considered baselines, see Table 5. We find that GFSVI also performs

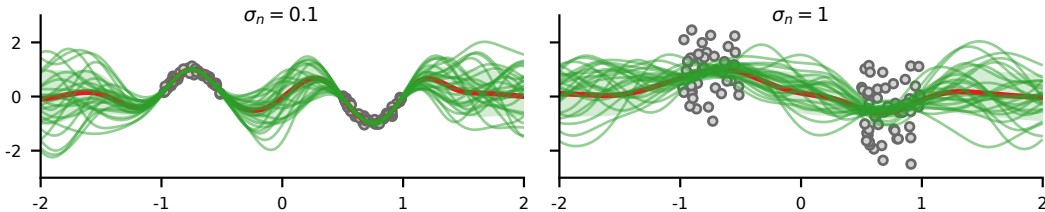

Figure 7: Our method (GFSVI) effectively regularizes functions generated by the Bayesian neural network (BNN) both in settings where the generative process is very noisy ($\sigma_n = 1$) or not ($\sigma_n = 0.1$).

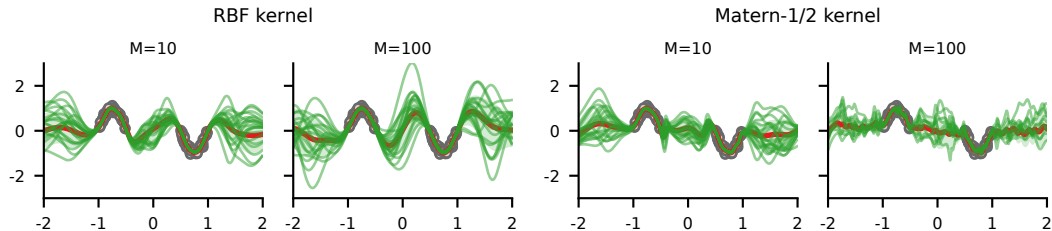

Figure 8: Our method (GFSVI) captures the smooth behavior of a Gaussian process (GP) prior with RBF covariance function even if the number of measurement points is small (N=10). However, in that setting GFSVI fails to reproduce the rough effect of a GP prior with a Matern-1/2 covariance function, and requires a larger amount of measurement points to do so (N=100).

Table 5: Test mean square error (MSE) of evaluated methods on regression datasets. We find that GFSVI (ours) also performs competitively in terms of MSE compared to baselines and obtains the best mean rank, matching best the performing methods on nearly all datasets.

| DATASET | FUNCTION SPACE PRIORS | | WEIGHT SPACE PRIORS | | | GOLD STANDARDS | |
|---|---|---|---|---|---|---|---|
| | GFSVI (OURS) | FVI | TFSVI | MFVI | LAPLACE | SPARSE GP | GP |
| BOSTON | $\mathbf{0.123 \pm 0.047}$ | $\mathbf{0.136 \pm 0.050}$ | $0.995 \pm 0.206$ | $0.532 \pm 0.160$ | $\mathbf{0.203 \pm 0.104}$ | $0.122 \pm 0.032$ | $0.115 \pm 0.045$ |
| CONCRETE | $\mathbf{0.114 \pm 0.019}$ | $\mathbf{0.116 \pm 0.009}$ | $0.389 \pm 0.035$ | $0.698 \pm 0.102$ | $\mathbf{0.116 \pm 0.015}$ | $0.399 \pm 0.045$ | $0.116 \pm 0.016$ |
| ENERGY | $0.003 \pm 0.000$ | $0.003 \pm 0.001$ | $0.003 \pm 0.001$ | $0.152 \pm 0.053$ | $\mathbf{0.002 \pm 0.000}$ | $0.087 \pm 0.011$ | $0.087 \pm 0.009$ |
| KIN8NM | $\mathbf{0.071 \pm 0.003}$ | $0.075 \pm 0.008$ | $0.073 \pm 0.003$ | $0.290 \pm 0.249$ | $0.083 \pm 0.003$ | $0.088 \pm 0.005$ | - |
| NAVAL | $\mathbf{0.000 \pm 0.000}$ | $0.001 \pm 0.001$ | $0.000 \pm 0.000$ | $0.007 \pm 0.006$ | $\mathbf{0.000 \pm 0.000}$ | $0.000 \pm 0.000$ | - |
| POWER | $\mathbf{0.052 \pm 0.002}$ | $0.054 \pm 0.005$ | $0.054 \pm 0.003$ | $0.058 \pm 0.004$ | $0.054 \pm 0.004$ | $0.071 \pm 0.003$ | - |
| PROTEIN | $0.459 \pm 0.011$ | $0.466 \pm 0.010$ | $\mathbf{0.429 \pm 0.008}$ | $0.537 \pm 0.017$ | $0.446 \pm 0.014$ | $0.408 \pm 0.003$ | - |
| WINE | $\mathbf{0.652 \pm 0.050}$ | $0.663 \pm 0.020$ | $1.297 \pm 0.208$ | $\mathbf{0.655 \pm 0.052}$ | $0.637 \pm 0.069$ | $0.607 \pm 0.074$ | $0.585 \pm 0.071$ |
| YACHT | $\mathbf{0.003 \pm 0.001}$ | $0.004 \pm 0.002$ | $0.221 \pm 0.082$ | $0.682 \pm 0.313$ | $\mathbf{0.002 \pm 0.001}$ | $0.399 \pm 0.144$ | $0.355 \pm 0.066$ |
| WAVE | $\mathbf{0.000 \pm 0.000}$ | $\mathbf{0.000 \pm 0.000}$ | $\mathbf{0.000 \pm 0.000}$ | $\mathbf{0.000 \pm 0.000}$ | $\mathbf{0.000 \pm 0.000}$ | $0.000 \pm 0.000$ | - |
| DENMARK | $\mathbf{0.155 \pm 0.009}$ | $0.287 \pm 0.007$ | $\mathbf{0.163 \pm 0.009}$ | $0.225 \pm 0.007$ | $0.194 \pm 0.007$ | $0.260 \pm 0.002$ | - |
| MEAN RANK | 1.182 | 1.545 | 1.545 | 2.182 | 1.182 | - | - |

competitively in terms of MSE compared to baselines and obtains the best mean rank, matching best the performing methods on nearly all datasets. In particular, we find that using GP priors in the linearized BNN setup with GFSVI yields improvements over the weight space priors used in TFSVI and that GFSVI performs slightly better than FVI. Function-space VI methods (TFSVI, GFSVI, FVI) significantly improves over weight space VI mostly performing similarly to the linearized Laplace approximation. Further improvement over baselines are obtained when considering GP priors with GFSVI and FVI. Finally, GFSVI compares favorably to the GP and sparse GP.

### A.3.3    ADDITIONAL PLOTS FOR KERNEL EIGENVALUE DECAY

The rate of decay of covariance operator's eigenvalues gives important information about the smoothness of stationary kernels (Williams & Rasmussen, 2006) and that increased smoothness of the kernel leads to faster decay of eigenvalues Santin & Schaback (2016). For instance, RBF covariance operator eigenvalues decay at near exponential rate independent of the underlying measure (Belkin, 2018) and Matern kernels eigenvalues decay polynomialy (Chen et al., 2021). We find that the kernel evaluated at points sampled from a uniform distribution over $\mathcal{X}$ share this same behavior (see Figure 9).

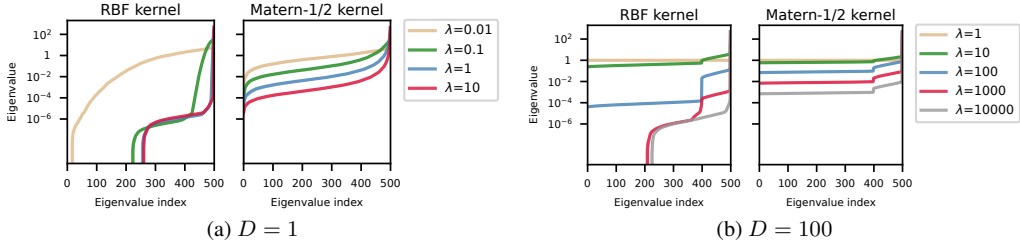

(a) $D = 1$        (b) $D = 100$

Figure 9: Mean eigenvalues of the Gram matrix obtained for different kernels and for varying length-scales over 10 draws from a uniform distribution on $[-2, 2]^D$. The mean eigenvalues are arranged in increasing order. The eigenvalues of the Gram matrix associated with the smooth RBF kernel decays much faster than those of the Matern-1/2. Furthermore, the eigenvalues decay at a slower rate in high dimensions (D=100).

### A.3.4 ADDITIONAL PLOTS FOR CHOOSING $\gamma$ IN $D_{\mathrm{KL}}^{\gamma}$.

The $\gamma$ parameter controls the magnitude of the regularized KL divergence (see Figure 12) and adjusts the relative weight of the regularized KL divergence and expected log-likelihood term in the training objective (see Figure 10). Furthermore, $\gamma$ also acts as "jitter" preventing numerical errors. We recommend choosing $\gamma$ large enough to avoid numerical errors while remaining small enough to provide strong regularization.

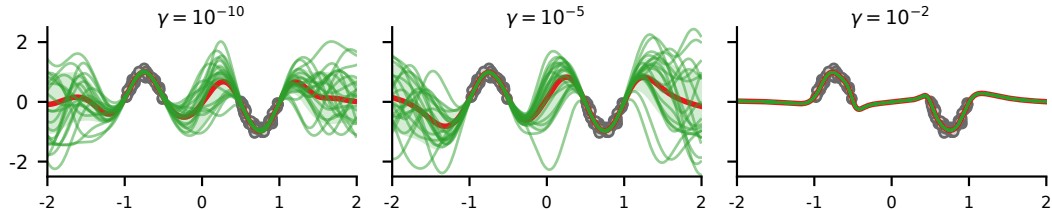

Figure 10: The $\gamma$ parameter of the regularized KL divergence controls the magnitude of the regularizer in the objective and should be small enough to provide strong regularization.

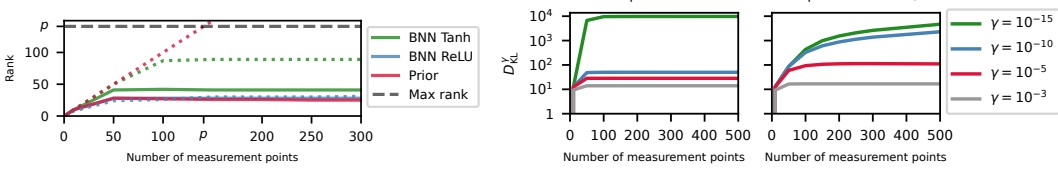

Figure 11: The BNN's covariance adaptation to the prior's covariance rank depends on its activation function. BNNs fit with a RBF prior (full) show lower rank than with a Matern-1/2 (dotted).

Figure 12: $\gamma$ explicitly controls the magnitude of the regularized KL-divergence $D_{\mathrm{KL}}^{\gamma}$. Rougher priors (Matern-1/2) require more measurement points to accurately estimate $D_{\mathrm{KL}}^{\gamma}$ than smooth priors (RBF).

