# OpenReview forum: "Regularized KL-Divergence for well-defined function space variational inference in BNNs"
_ICLR.cc/2024/Conference — Submitted to ICLR 2024_

### Official Review · Reviewer_izSH · 2023-10-29

**Soundness:** 3 good
**Presentation:** 3 good
**Contribution:** 2 fair
**Rating:** 3
**Confidence:** 3

**Summary:**

This paper proposes a new function space variational inference for BNNs based on generalized KL divergence. The framework follows the linearized functional BNNs (Rudner et al. 2022). The new objective function is claimed to be a well-defined objective compared with the original possible ill-defined objective function. The experimental results show good performance.

**Strengths:**

The paper is well presented with good organization. The issues of existing functional VI for BNNs are well described, and the problem of this paper is well-motivated. Sufficient background is given to understand the problem and the proposed idea. The proposed method is simple and looks good from the experiments.

**Weaknesses:**

The authors claim that ‘VI is too restrictive for BNNs with informative function space priors’ is one of the contributions of this paper. However, the discussions in Section 2.2 just follow the existing works and the proofs in Appendix 1.1 are also a repeat of existing works. This point as a contribution is too weak.

Although the idea of introducing generalized KL divergence may be reasonable, the final objective is without much difference from the previous one. During the practical implementation of previous functional BNNs, we usually add a diagonal identify matrix to ensure the non-singular the covariance matrix in (2) or (11). Hence, the technique contribution is weak. In the experiments, there are no results to show the cases where the non-singular covariance matrix impacts greatly the results.

**Questions:**

Please see the Weaknesses.

---

> ### Author Response · Authors · 2023-11-17
>
> We thank the reviewer for their time and informative feedback. We are encouraged that they found our method well motivated, simple and that it provides good experimental results. We are also glad they thought the background and issues with function-space BNNs were well presented.
>
> The review has led us to realize the need for some clarifications in the paper, and we will upload a revised version in the coming days. In the meantime, please find our answers and comments below.
>
> > Although the idea of introducing generalized KL divergence may be reasonable, the final objective is without much difference from the previous one.
>
> We realize that the differences to previous work require some clarification in our paper. We invite the reviewer to read items 1 and 3 in the overall response (“Importance of function-space priors and fundamental differences to TFSVI and FVI“) which highlights how our method differs from its closest baselines TFSVI (Rudner et al., 2022) and FVI (Sun et al., 2019).
>
> In brief, our method allows for informative priors in *function space* whereas TFSVI requires priors to be specified in weight space. This may not be obvious since TFSVI pushes the weight-space prior forward into function space, but the choice of priors in TFSVI is still limited to distributions in weight space where meaningful belief specification is difficult e.g. see Figure 4 in the appendix. For details, see item 1 in the overall response.
>
> Compared to FVI, our method uses the linearized BNN and avoids score function estimators (see Issue (i) in Section 2.2), which have been reported to make FVI difficult to use in practice (Ma et Hernández-Lobato 2021). From a more theoretical point of view, we show that, within the linearized BNN, the KL-divergence estimator of FVI (which introduces noise in a somewhat ad-hoc way) can be well-motivated and interpreted as an estimator of a well-defined regularized KL-divergence. See item 3 in our overall response for more details.
>
> > During the practical implementation of previous functional BNNs, we usually add a diagonal identity matrix to ensure the non-singular the covariance matrix in (2) or (11)
>
> We agree that existing methods also introduce such jitter, and that it is somewhat subtle how the role of jitter differs across methods. We invite the reviewer to read items 2 and 3 in our overall response, which discusses these differences between our method, TFSVI, and FVI.
>
> In summary, the jitter introduced in the implementation of TFSVI only addresses numerical issues that would not arise in the absence of rounding errors (see Issue (iii) in Section 2.2). By contrast, the jitter in our method does not solve numerical issues but instead is fundamentally necessary as we calculate a divergence between measures with different support, which would be infinite without the jitter.
>
> FVI also adds jitter to the variational posterior and prior when estimating the KL divergence. This jitter serves a similar purpose as in our method. However, since FVI does not linearize the BNN, it does not have access to an explicit variational measure in function space. Therefore, in order to take jitter into account, FVI has to draw Gaussian noise. By contrast in the linearized BNN, the marginalization over jitter can be done analytically as the variational posterior is a Gaussian measure.
>
>
> > In the experiments, there are no results to show the cases where the non-singular covariance matrix impacts greatly the results.
>
> We are not sure to understand the reviewer’s question. If the reviewer asks why we present results for TFSVI and FVI despite explaining that these methods do not work without jitter (see Section 2.2 issue (iii)), it is because our implementations are based on the code provided by their respective authors which add jitter in the KL divergence estimators. If the reviewer asks for additional experiments where we use our method, TFSVI or FVI without jitter, then we would like to point out that training such models would be impossible as we would immediately run into singularities.
>
> Our experiments show that our method is better than FVI and TFSVI at capturing prior beliefs (see, e.g., Figure 2, and Figure 4 in the appendix), and that it leads to better predictive performance and out-of-distribution detection (Tables 1 and 3).
>
> > The authors claim that “VI is too restrictive for BNNs with informative function space priors”
>
> After consideration, we agree with the reviewer that this point is a bit weak as a contribution. Indeed, our argument for this claim builds entirely on the results by Burt et al. (2020). We will remove this claim for the paper. The main contribution of our paper is not theoretical, but instead of a practical and well-defined method that empirically performs well.
>
> We hope that our response clarifies how our contribution differs from prior work, and that we were able to convince you that our simple method could be helpful to members of the community.

---

> > ### Comment · Reviewer_izSH · 2023-11-22
> >
> > Thanks for the authors' responses!
> >
> > I am still concerned about the difference between the TFSVI and the proposed method. The objective function of the proposed method is just with a "jitter" term compared with previous work and such a term is usually added for any implementation of GP-related algorithms. Hence, I will keep my score.

---

> > > ### Author Response · Authors · 2023-11-22
> > > **Fundamental difference between TFSVI and our method**
> > >
> > > Thank you for the quick response. To clarify, we would like to stress that TFSVI is fundamentally different to our method because TFSVI is restricted to prior specification in weight space. This prohibits TFSVI from introducing any form of meaningful prior knowledge into BNNs.
> > >
> > > We invite the reviewer to consider Figure 1 in our paper, where we perform BNN inference with various different function-space priors, and we show that our method successfully adapts to the different smoothness assumptions encoded in these priors. The reason why we do not show corresponding plots for TFSVI in this figure is because these experiments would be fundamentally impossible to do with TFSVI (because the method does not have any function space prior to begin with, and finding equivalent weight-space priors for different function space priors is a difficult problem in itself). In response to reviewer feedback, our updated version of the paper contains a paragraph on "Differences to prior work" at the end of Section 3.3, which reiterates that TFSVI does not admit prior specification in function space.

---

### Official Review · Reviewer_PriF · 2023-11-01

**Soundness:** 3 good
**Presentation:** 3 good
**Contribution:** 1 poor
**Rating:** 3
**Confidence:** 4

**Summary:**

This paper proposes a generalized function space variational inference (GFSVI) for Bayesian neural networks (BNNs), where the intractable and often ill-defined function-space KL term is replaced with a regularized KL divergence. While the original KL divergence in function space can in principle blow up to infinity, and this actually happens for many practical applications, the regularized KL divergence does not suffer from the same issue, thus providing more stable results. The proposed method has been demonstrated on synthetic and real-world regression tasks.

**Strengths:**

- The paper is well-written and easy to follow.
- Bringing regularized KL to function space variational inference is a good contribution that might benefit the community.
- Good empirical results on UCI regression benchmarks.

**Weaknesses:**

- No significant contribution. As far as I could understand, the only contribution of this paper is to bring the regularized KL divergence which was well developed by (Quang 2019), and use it as a substitute for the ordinary KL divergence in the framework of tractable function space VI (TFVI, Rudner et al., 2022). Other than this, I fail to see any contribution, and even the combination of those two methods is implemented quite straightforwardly, without any issue to consider during that process.
- Limited experiments. While I appreciate the experiments on the regression tasks, they are relatively small-scale tasks, and only the small BNNs (MLPs mostly) are tested. It is hard to judge the effectiveness of the proposed method without scaling, for instance, the image classification task (CIFAR-10 or CIFAR-100, at least) solved with ResNet, as typically done in the literature.

**Questions:**

- Once we replace the KL divergence with regularized KL divergence, the resulting objective becomes something that is different from ELBO, so we end up optimizing an objective that is not particularly a lower bound on the marginal likelihood. There is a class of inference algorithms (generalized Bayesian inference) generalizing the standard Bayesian inference procedure and thus variational inference by extending the likelihood or KL regularization term with more flexible functions, and for such cases the utility of the extended objective can be described in the perspective of generalizaton error. However, in the current form, the alternative objective with the regularized KL does not explain anything about its optimum. I think there should be some intuitive justification for this.

---

> ### Author Response · Authors · 2023-11-17
>
> We thank the reviewer for their time and informative feedback. We are encouraged that they thought that the regularized KL divergence was a good contribution for the community and that they found our method to perform well on regression tasks. We are also glad they thought the paper to be well written and easy to follow.
>
> The review has led us to realize the need for some clarifications in the paper that we will make in the coming days. In the meantime, please find our answers and comments below.
>
> > the only contribution [...] is to bring the regularized KL divergence [...] as a substitute for the [...] KL divergence in the framework of [...] TFVI (Rudner et al., 2022).
>
> We kindly refer to the reviewer to item 1 and 3 which highlights how our method differs from TFSVI and FVI (Sun et al., 2019) in the overall response (“Importance of function-space priors and fundamental differences to TFSVI and FVI“).
>
> In summary, our main contribution is a simple method for inference in BNNs with *function-space priors*. Unlike our method, TFSVI defines prior beliefs in weight space using standard BNN priors, but pushes it to function-space to compute the KL divergence. Thus priors in TFSVI are still limited to distributions in weight space where meaningful belief specification is difficult e.g. see Figure 4 in the appendix. For details, see item 1 in the overall response.
>
> Compared to FVI, our method uses the linearized BNN and avoids score function estimators (see Issue (i) in Section 2.2), which have been reported to make FVI difficult to use (Ma et Hernández-Lobato 2021). From a more theoretical point of view, we show that, within the linearized BNN, the KL-divergence estimator of FVI (which introduces noise in a somewhat ad-hoc way) can be well-motivated and interpreted as an estimator of a well-defined regularized KL-divergence. See item 3 in our overall response for more details.
>
> > While I appreciate the experiments on the regression tasks, they are relatively small-scale tasks, and only the small BNNs (MLPs mostly) are tested.
>
> Indeed, the BNNs we consider are relatively small as is typical in the literature on function space VI for BNNs¹. Computing the regularized KL divergence can be expensive for large models. Thus, in its current state, we believe that our method is best suited for models in a regime in-between analytically solvable and deep learning models, which find many practical applications involving decision making, such as Bayesian optimization or bandits. At the same time, we believe that one of the strengths of our method is its simplicity, which would make it a solid starting point for future research, including ways to make it more scalable.
>
> ¹ note that Rudner et al. (2022) reports results using a Resnet-18 on a Nvidia V-100 GPU with 32GB of memory. It should be possible to fit this model with our method using similar resources. However our method offers little advantages for classification tasks, as discussed in the next answer below.
>
> > It is hard to judge the effectiveness of the proposed method without [...] the image classification task (CIFAR-10 or CIFAR-100, at least) solved with ResNet, as typically done in the literature.
>
> We thank the reviewer for raising this point on which we did not expand on in the paper. Our main contribution is the use of GP priors for BNNs to incorporate *informative knowledge* about the functions generated by the BNN. Prior beliefs in the classification setting are defined in logit-space before the output activation which are much harder to specify and puts into doubt our original motivation. This is why we preferred to restrict our method to regression tasks. However, automatic prior selection methods with our method might be beneficial for classification and would be an interesting direction for future work.
>
> > There is a class of inference algorithms (generalized Bayesian inference) [...] can be described in the perspective of generalization error. However, [...] the [...] objective with the regularized KL does not explain anything about its optimum. I think there should be some intuitive justification for this.
>
> Thank you for raising this point. We are not sure if we understand the question. As discussed at the beginning of Section 3, our method builds on generalized VI (Kloblauch et al., 2018), which views Bayesian inference as regularized empirical risk minimization by considering the ELBO as a regularized expected log-likelihood objective. While we summarize the main idea behind generalized VI at the beginning of Section 3, we think that adding more details about generalized VI would be beyond the scope of our paper, and we instead refer readers to the original paper. We hope this answers the question and we are happy to follow up if we misunderstood it.
>
> We hope that our response clarifies how our contribution differs from prior work, and that we were able to convince you that our simple method could be helpful to members of the community.

---

> > ### Comment · Reviewer_PriF · 2023-11-21
> > **Response to the rebuttal**
> >
> > I appreciate the authors' effort to clarify the issues raised in my reviews. I apologize for my misunderstanding on generalized VI; I missed that the paper justified the objective function using that.
> >
> > However, I'm still not convinced by the claim on the novelty and the lack of larger-scale experiments; my main point (and I think the other reviewers are with me on this point) is that the presented paper is a combination of existing methods without significant technical innovation. Also, while I agree that some aspects of the proposed method can be highlighted through the experiments presented in this paper, without demonstration for practicality (at least an image classification benchmark with moderate-sized deep neural nets that do not require large memories), it is hard to value the contribution of the paper to the community (especially considering the lack of novelty claim). So I decided to keep my original score.

---

### Official Review · Reviewer_sJLb · 2023-11-01

**Soundness:** 3 good
**Presentation:** 3 good
**Contribution:** 2 fair
**Rating:** 5
**Confidence:** 3

**Summary:**

This work focuses on a challenge in function-space variational inference, where the KL divergence between two stochastic processes, evaluated on a finite number of inputs, could have an infinite value. This issue raises the numerical issue when training the Bayesian Neural Networks (BNNs) by function-space variational inference (VI). To address this problem, the authors employ the regularized KL divergence, which is defined to have a finite value and can be used to resolve the mentioned issue. Empirically, the authors demonstrate that function-space VI using the regularized KL divergence leads to better uncertainty estimation on synthetic and UCI regression datasets.

**Strengths:**

### Justification for jitter addition
> It seems that this work justifies the addition of jitter to each covariance term to address the numerical issue, i.e., the infinite value of function-space KL divergence by introducing the well-established regularized KL divergence into the framework of the function space VI.

**Weaknesses:**

### Incremental contribution

> Compared to the tractable function space VI of [1], it seems that the objective in Eq. (9) exhibits only minor differences, i.e., jitter term ($\gamma M I_M$) in each covariance term in Eq. (11). Based on my understanding, adding the jitter term has been commonly used in implementation to handle the numerical issue when training the model with the Gaussian KL divergence as KL objective. Therefore, the proposed objective itself does not seem novel in sense of training objective for VI.


### Experiment results are limited to regression setting.

> While the tractable function space VI of [1] has been demonstrated on both classification and regression tasks, this work has been demonstrated only for regression experiment setting.

[1] Tractable Function-Space Variational Inference in Bayesian Neural Networks - NeurIPS 22

**Questions:**

*  I could not identify significant differences between the tractable function space VI of [1] and the proposed method. What is the primary distinction in comparison to [1]?

* In comparison to [1], what specific difference in the proposed method leads to the improved performance in Table 1?

* Regarding Table 1, why is TFSVI categorized under weight space priors? As far as I understand, the KL divergence of TFSVI is evaluated in the function space using the push-forward distribution of the weight parameter distribution, which is defined in the function space.


[1] Tractable Function-Space Variational Inference in Bayesian Neural Networks - NeurIPS 22

---

> ### Author Response · Authors · 2023-11-17
>
> We thank the reviewer for their useful and informative feedback on our paper. Please find our responses below.
>
> > - “I could not identify significant differences between the tractable function space VI of [1] and the proposed method. What is the primary distinction in comparison to [1]?”
> > - “Why is TFSVI categorized under weight space priors?”
> > - “In comparison to [1], what specific difference in the proposed method leads to the improved performance in Table 1?”
>
> We kindly refer the reviewer to point 1 in our overall response (“Importance of function-space priors and fundamental differences to TFSVI and FVI”). In brief, our method allows specifying prior beliefs in function space whereas TFSVI specifies the prior in weight space, but pushes it forward to function space to calculate the KL-divergence. However, such a push-forward only changes the *representation* of the prior and does not lift the limitations of expressing meaningful prior beliefs in weight-space since neural networks weights are not interpretable.
> These differences explain the improved performance of our method in Tables 1 and 3, and the qualitative improvements in Figure 2, and Figure 4 in the appendix. In these experiments, we chose the best weight-space prior that we could find for TFSVI by cross validation.
> > “jitter term has been commonly used in implementation to handle the numerical issue when training the model with the Gaussian KL divergence as KL objective”
>
> Thank you for bringing this up. The different ways in which TFSVI, FVI (Sun et al., 2019), and our method introduce jitter is indeed somewhat subtle, but these differences are crucial. We discuss the differences in paragraph issue (iii) of Section 2.2, but we realize based on reviewer feedback that this issue requires further explanation. We kindly refer to points 2 and 3 in our overall response above, which we hope clarify these differences (we will add a similar clarification to the paper over the next few days). We provide a synopsis below.
>
> TFSVI introduces jitter to avoid numerical issues. Since TFSVI defines both prior and variational distributions in weight space, their respective push-forwards into function space have the same support, and the KL-divergence between them would be technically finite if it wasn’t for numerical errors. By contrast, the jitter in our method does not resolve numerical issues but instead is fundamental to obtaining a finite objective. Indeed, in our method the prior and variational posterior have different support since the prior is specified in function space while the variational posterior is still limited to the push-forward of a variational distribution in weight space.
>
> Compared to our method, FVI does not linearize the BNN and only *implicitly* defines the variational measure in function-space which makes the method substantially more complicated to use in practice (Ma et Hernández-Lobato 2021). Thus, “adding jitter” in FVI means drawing actual realizations of Gaussian noise. By contrast, we use the linearized BNN which yields a Gaussian variational posterior for which we can marginalize over the jitter analytically. It turns out that applying the estimator of FVI to our situation is equivalent to calculating the estimator of the regularized KL-divergence, which has an *explicit* form that we can directly optimize.
>
> > “Compared to the tractable function space VI of [1] (TFSVI), it seems that the objective in Eq. (9) exhibits only minor differences”
>
>
> We agree that the *implementation* of our method is similar to TFSVI. But we see this as a strength rather than a shortcoming. In fact, we believe that our method has the potential of becoming a go-to solution for function-space BNNs precisely because it can be implemented easily and yet evidently offers improvements over TFSVI and FVI.
>
> > “While the tractable function space VI of [1] has been demonstrated on both classification and regression tasks, this work has been demonstrated only for regression experiment setting.”
>
> We thank the reviewer for raising this point that we did not clarify in the paper. As mentioned above, the main difference between our method and the one in [1] is the use of GP priors in function space, which allow us to incorporate knowledge about the properties of the functions generated by the BNN. Prior beliefs in the classification setting are specified in the space of logits before the output activation, which is less interpretable than for regression and puts into question our initial motivation for informative priors. For this reason, we prefer to concentrate on the regression setting. However, using automatic prior selection along with our method might provide benefits over standard BNNs in classification tasks and would be an interesting direction for future work.
>
> We hope that our response clarifies how our contribution differs from prior work, and that we were able to convince you that our simple method could be helpful to members of the community.

---

> ### Comment · Reviewer_sJLb · 2023-11-23
> **Official Comment by Reviewer sJLb**
>
> Thank you for responding to my questions. Your responses have helped me better understand the distinction between the proposed method and TFSVI, as claimed by the authors. However, I am somewhat skeptical that using Gaussian processes to construct a function-space prior can result in an interpretable function-space prior for a given dataset. Based on my understanding, the inductive bias of the GP prior could be interpretable for certain types of covariance functions like the RBF kernel, Matern, and Periodic kernel. Choosing the proper kernel function for GP seems non-trivial and depends on the dataset. Thus, I believe that relying on a GP prior to build an interpretable function-space prior might still have some limitations.
>
> Personally, I think that providing a specific method for forming the interpretable function-space prior would further enhance the novelty of this work, especially considering the distinction between the proposed method and TFSVI.
>
> With these reasons, I will maintain my score for the current content of the manuscript.

---

### Author Response · Authors · 2023-11-16
**Overall response to reviews: Importance of function-space priors and fundamental differences to TFSVI and FVI**

We thank all reviewers for their time and for their valuable feedback. We will reply to each reviewer individually over the next few days. Before we do so, we would like to clarify the fundamental contribution of our paper to all reviewers, as we realized that our presentation left room for misunderstandings. We will upload a new version that clarifies the three points below.

The main contribution of our paper is a simple and well-defined variational method for performing inference in Bayesian neural networks (BNNs) using *priors in function space*. This distinguishes our method from TFSVI as we clarify in point (1) below. Further, we clarify in points (2) and (3) below that the jitter introduced in our method is different to the jitter introduced in TFSVI, and that our method deals with it in a more tractable way than FVI.

**(1) Difference to TFSVI:** in contrast to our proposed method, TFSVI (Rudner et al., 2022) defines prior beliefs in *weight space*. This may not be obvious as TFSVI pushes the prior forward to function space to calculate the KL-divergence. But as the prior specification happens in weight space, TFSVI still suffers from the same difficulties regarding prior specification as mean-field VI (Blundell et al., 2015), Laplace (Mackay, 1992), or any other weight-space BNN method. This can be seen, e.g., in Figure 4 in the appendix, where our method accurately captures the roughness of the (function-space) Matern-1/2 prior, but we were unable to find a corresponding weight-space prior for TFSVI that would replicate this behavior.

**(2) Jitter in TFSVI:** some reviewers have correctly pointed out that the implementation of TFSVI also introduces jitter. However, the jitter in TFSVI serves a very different purpose than in our objective: since TFSVI uses a weight-space prior, its push forward to function space has the same support as the variational posterior, and thus the KL-divergence is technically finite (Burt et al., 2020) if it weren’t for numerical errors (see penultimate paragraph of Section 2.2). By contrast, the function-space prior used by our method typically has a different support than the variational posterior, and thus the jitter (i.e., regularization) is necessary independent of numerical considerations.

**(3) Difference to FVI:** our method can be seen as an approximation of FVI (Sun et al., 2019) that makes it more tractable and thereby more effective in practice (see our empirical results). Similar to the regularized KL-divergence used by our method, FVI adds noise to both the prior and the variational posterior. While the original FVI-paper motivated the noise by numerical stability, we can now motivate it with the hindsight of Burt et al. (2020), who show that the function-space KL divergence would be infinite without the noise. However, our method handles the noise in a fundamentally different way than FVI. FVI does not have access to an explicit variational measure in function space because it does not linearize the BNN. This severely complicates the estimation of (gradients of) the KL divergence in FVI, and the authors resort to implicit score function estimators, which make FVI difficult to employ in practice (Ma et Hernández-Lobato, 2021). Our method does not suffer from these complications as linearizing the BNN allows us to marginalize over the noise analytically (Eqs. 11 and 8). Note that this marginalization is only possible in the *estimator* of the regularized KL divergence (Eq. 7), which is defined in finite dimensions, and not in the regularized KL divergence itself (Eq. 6), which operates on measures in infinite-dimensional spaces (adding white noise directly to a GP would violate the conditions for Gaussian measures, see Section 3.1). These soundness issues are why we introduced our method from the perspective of a regularized divergence, but we realize now that a more pragmatic explanation as the one in this paragraph is also useful, and we will add it to the paper.

We thank the reviewers again for their useful comments. We will upload a new version that includes the above clarifications, and we will respond to each reviewer’s individual questions in the coming days.

---

### Author Response · Authors · 2023-11-23
**Summary of updates to the paper and of our responses to reviewers**

As the public discussion period is about to end, we would like to thank the reviewers again for their time. As promised in our responses, we updated the paper to clarify the points that were raised by the reviewers:

- **Clarifying our contribution with respect to TFSVI and FVI:** we added a new paragraph in Section 3.3 (“Differences to prior work”) highlighting the differences between TFSVI (Rudner, 2022), FVI (Sun et al., 2019) and our method (see also items (1) and (3) of our overall comment here on openreview, as well as additional smaller clarifications in the 2nd paragraph of Section 1 and the 1st paragraph of Section 5 of our paper).

- **Role of jitter in TFSVI, FVI and our method:** we added an explanation to the paragraph “Issue (iii)” of Section 2.2 and to the new paragraph “Differences to prior work” in Section 3.3, which highlight how jitter in TFSVI serves a different purpose than in our method, and that jitter in FVI leads to practical difficulties (see also items (2) and (3) of our overall comment below).

- **No experiments in the classification setting:** we clarified at the beginning of Section 4 that prior beliefs in this setting are harder to specify than for regression, which would question our motivation based on better priors.

- **Change listed contributions:** we reformulated the list of contributions at the end of Section 1 to avoid insinuating that we are the first to highlight that the ELBO is not an appropriate objective for inference in BNNs with function space priors.

We finally wish to highlight the importance of function-space priors in BNNs, as this appeared to be a source of confusion. It is well documented in the BNN literature that the choice of prior significantly affects performance. Unfortunately it is difficult to find informative priors when one has to specify them in weight space. Our paper proposes a principled and simple method for performing inference in BNNs with informative (GP) priors in *function space*, and we observe significant qualitative and quantitative performance improvements over existing weight-space and function-space inference methods for BNNs (see Table 1, 2, 3 and Figures 2 and 4 in Appendix).

We hope that the reviewers find the updated paper clearer and that they agree that it now provides a stronger submission.

---

### Meta-Review · Area_Chair_iDw2 · 2023-12-13

**Metareview:**

This paper introduces a new objective for function-space variational inference in Bayesian neural networks, motivated by a generalized variational inference view. This leads to a slightly different formulation of the objective than in existing work, and allows for direct specification of function-space priors. However, reviewers felt that the paper did not clearly present its contribution, feeling the paper is too straightforward a combination of different existing work in its current form for ICLR; there were also requests for a broader set of experiments.

**Justification For Why Not Higher Score:**

Issues identified by reviewers include an overall limited contribution; and experiments which are small-scale and only on regression.

**Justification For Why Not Lower Score:**

N/A

---

### Decision · Program_Chairs · 2024-01-16

Reject